# hsa_circ_0001275 Is One of a Number of circRNAs Dysregulated in Enzalutamide Resistant Prostate Cancer and Confers Enzalutamide Resistance In Vitro

**DOI:** 10.3390/cancers13246383

**Published:** 2021-12-20

**Authors:** Marvin C. J. Lim, Anne-Marie Baird, John Greene, Ciara McNevin, Karine Ronan, Petar Podlesniy, Orla Sheils, Steven G. Gray, Ray S. McDermott, Stephen P. Finn

**Affiliations:** 1Department of Histopathology and Morbid Anatomy, Trinity Translational Medicine Institute, Trinity College Dublin, D08 W9RT Dublin, Ireland; greenejo@tcd.ie (J.G.); csmcnevin@gmail.com (C.M.); osheils@tcd.ie (O.S.); 2Department of Medical Oncology, St. James’s Hospital, D08 NHY1 Dublin, Ireland; 3Department of Medical Oncology, Tallaght University Hospital, D24 NR0A Dublin, Ireland; ray.mcdermott@tuh.ie; 4School of Medicine, Trinity Translational Medicine Institute, Trinity College Dublin, D08 W9RT Dublin, Ireland; bairda@tcd.ie (A.-M.B.); sgray@stjames.ie (S.G.G.); 5Department of Medical Oncology, St. Vincent’s University Hospital, D04 YN26 Dublin, Ireland; karine.ronan@ucdconnect.ie; 6CiberNed (Centro Investigacion Biomedica en Red Enfermedades Neurodegenerativas), IIBB, Rosello 161, 08036 Barcelona, Spain; petar_podlesniy@bio-rad.com; 7Thoracic Oncology Research Group, Labmed Directorate, St. James’s Hospital, D08 RXOX Dublin, Ireland; 8School of Biological and Health Sciences, Technological University Dublin, D07 ADY7 Dublin, Ireland; 9Department of Histopathology, St. James’s Hospital, D08 X4RX Dublin, Ireland

**Keywords:** prostate cancer, enzalutamide resistance, circRNA (circular RNA)

## Abstract

**Simple Summary:**

Although newer generations of androgen deprivation therapy such as enzalutamide are providing hope, it is clinically challenging to deliver effective therapy to individuals with metastatic castrate-resistant prostate cancer. Between 20–40% of patients have intrinsic resistance to therapy and all patients will ultimately experience disease progression due to acquired resistance, which is a significant clinical dilemma. The aim of our study was to evaluate the role of circular RNAs (circRNAs) in enzalutamide-resistant prostate cancer as part of the effort to identify useful biomarkers for patient selection and potential new therapeutic targets. We confirmed that hsa_circ_0001275 was highly upregulated in an enzalutamide resistant cell line and demonstrated that its overexpression resulted in increased enzalutamide resistance. Our data showed that hsa_circ_0001275 was not expressed abundantly in patient plasma samples, however, a trend of expression was evident which paralleled disease activity indicating a possible association with enzalutamide resistance. Overall, we have provided evidence that hsa_circ_0001275 promotes enzalutamide resistance and thus may serve as a potential therapeutic target.

**Abstract:**

Background: Enzalutamide is part of the treatment regimen for metastatic castration-resistant prostate cancer (MCRPC). However, both intrinsic and acquired resistance to the drug remain substantial clinical quandaries. circRNAs, a novel type of non-coding RNA, have been identified in a number of cancers including prostate cancer and have been associated with cancer development and progression. circRNAs have shown great potential as clinically useful blood-based ‘liquid biopsies’ and as therapeutic targets in prostate cancer. The aim of this study was to examine the role of circRNA transcripts in enzalutamide-resistant prostate cancer cells and assess their utility as biomarkers. Methods: An isogenic cell line model of enzalutamide resistance was subjected to circRNA microarray profiling. Several differentially expressed circRNAs, along with their putative parental genes were validated using reverse transcription-quantitative polymerase chain reaction (RT-qPCR). circRNAs of interest were stably overexpressed in the control cell line and drug sensitivity was assessed using an ELISA-based proliferation assay. The candidate circRNA, hsa_circ_0001275, was measured in patient plasma samples using RT-droplet digital PCR (RT-ddPCR). Results: hsa_circ_0001275 and its parental gene, PLCL2, were significantly up-regulated in strongly resistant clones vs. control (*p* < 0.05). Overexpression of hsa_circ_0001275 in the control cell line resulted in increased resistance to enzalutamide (*p* < 0.05). While RT-ddPCR analysis of hsa_circ_0001275 expression in plasma samples of 44 clinical trial participants showed a trend that mirrored the stages of disease activity (as defined by PSA level), the association did not reach statistical significance. Conclusions: Our data suggest that increased levels of hsa_circ_0001275 contribute to enzalutamide resistance. hsa_circ_0001275 plasma expression showed a trend that mirrors the PSA level at specific disease time points, indicating that circRNAs mirror disease recurrence and burden and may be associated with enzalutamide resistance.

## 1. Introduction

Prostate cancer is the most common invasive cancer diagnosed in men and the second leading cause of male cancer-associated mortality in the United States (US) [1]. The majority (79%) of patients with prostate cancer are diagnosed at a localised stage [2], which is potentially curable using radical prostatectomy, external beam radiotherapy (EBRT), or brachytherapy [3]. However, about 35% to 60% of men undergoing the initial curative effort will experience disease recurrence [4,5,6]. For these patients, androgen deprivation therapy (ADT) is the standard of treatment [7] but resistance inevitably develops resulting in castration-resistant prostate cancer (CRPC), which is currently incurable [8].

The treatment strategy for prostate cancer has evolved rapidly in the last decade due to greater availability and selection of therapeutic agents that improve overall survival (OS) of patients with CRPC such as second-generation antiandrogens (abiraterone and enzalutamide), chemotherapy (cabazitaxel), and radionuclide therapy (alpha-radium 223) [9,10,11,12]. The second-generation anti-androgens have stood at the forefront of this progress, however, despite the effectiveness of these new agents, 20% to 40% of patients will have intrinsic resistance to these therapies and all patients will eventually experience disease progression due to acquired resistance [9,13,14]. Additionally, recent phase III studies such as CHAARTED and LATITUDE have highlighted the importance of appropriate sequencing of treatment in optimizing clinical benefit and outcomes for patients with metastatic castrate sensitive prostate cancer (MCSPC) [15,16]. Even though an impressive therapeutic armamentarium for prostate cancer exists, there is a lack of effective clinical tools centred on patient selection making it difficult to provide the optimal therapy at each time point along the care pathway. 

Our knowledge of the genomic and transcriptomic landscape of prostate and other cancers has broadened substantially due to improved sequencing technologies, in turn leading to the ability to study previously poorly defined biomarkers such as non-coding RNA (ncRNA) contributing to novel molecular biomarker discoveries and new therapeutic strategies [17,18]. ncRNA consists of several different classes including circular RNA (circRNA), microRNA (miRNA) and long ncRNA (lncRNA) [19]. circRNA is produced from the backsplicing of exons, introns, or both where the 3′ and 5′ ends are covalently closed into one continuous loop, which critically confers resistance to degradation by RNA exonucleases [20,21]. Thus, circRNA is present in the cytoplasm owing to its stability and can be partitioned into exosomes [22,23]. For these reasons, circRNAs are believed to be a novel stable plasma biomarker. Importantly, the triad of stability, abundance and evolutionary preservation between species indicate that circRNA may have a substantial regulatory function [24] and appear to have essential roles in cancer initiation, development and progression [25,26]. A number of circRNA biomarkers have been investigated in prostate cancer as summarised by Papatsirou et al. [27], however, there are few studies in the literature describing the role of circRNA in enzalutamide resistance in vitro [28,29]. For instance, Wu et al. demonstrated that the downregulation of circRNA17 resulted in the overexpression of AR-V7 leading to enzalutamide resistance in prostate cancer cell lines [28] and Greene et al. found that the downregulation of hsa_circ_0004870 may lead to enzalutamide resistance through the regulation of AR-V7 [29]. Enzalutamide competitively binds to the carboxy-terminal ligand domain (LBD) of AR [9], thus hindering AR translocation, AR cofactor recruitment, and AR binding to DNA [30]. Enzalutamide has been shown in previous phase 3 studies to prolong OS and progression-free survival (PFS) in patients who were chemotherapy naive [31] and in the cohort of patients who had chemotherapy [9]. 

Our group previously identified differential circRNAs expression in an enzalutamide resistance model, which consisted of age-matched LNCaP parental cells (Control) and two sub-lines displaying strong resistance (Clone 1) and weak resistance (Clone 9) [29]. In this current study, two highly expressed circRNAs (hsa_circ_0001275 and hsa_circ_0001721) in Clone 1 were investigated in vitro, and expression levels determined using RT-ddPCR in plasma samples of patients with MCRPC collected as part of the Phase II trial of radium-223 in combination with enzalutamide (CTRIAL-IE 13-21, NCT02225704) [32].

## 2. Materials and Methods

### 2.1. Cell Lines

Isogenic LNCaP clones were generated from single clones as described by Korpal et al. [33], which consisted of aged-matched LNCaP parental cells (Control) and two sub-lines displaying strong resistance (Clone 1) and weak resistance (Clone 9) to enzalutamide. These clones were gifted from Novartis (Dublin, Ireland). All clones were cultured in RPMI-1640 (Sigma-Aldrich, St. Louis, MO, USA) supplemented with 10% foetal bovine serum (FBS) (Sigma-Aldrich, St. Louis, MO, USA) and 1% penicillin, streptomycin, and amphotericin b (Sigma-Aldrich, St. Louis, MO, USA) at 37 °C in a humidified atmosphere containing 5% CO_2_. All cell lines were periodically examined for mycoplasma infection and experiments were undertaken at passages below 30. The STR profiles for the LNCaP Control, C1 and C9 cells were compared against the ATCC STR database (https://www.atcc.org/search-str-database, accessed on 19 November 2021). The matching criterion is based on an algorithm that compares the number of shared alleles between two cell line samples, expressed as a percentage. Cell lines with ≥80% match are considered to be related; derived from a common ancestor. Using the following STR criteria: (AMEL) + the following loci: D5S818, D13S317, D7S820, D16S539, vWA, TH01, TPOX, CSF1PO. The results for identity using the ATCC algorithm were as follows: LNCAP Control: 94% (Pass), LNCaP C1: 94% (Pass), LNCaP C9: 94% (Pass).

### 2.2. Total RNA Preparation and Reverse Transcription

The miRNeasy Mini Kit (Qiagen, Manchester, UK) was used to isolate RNA from cell lines according to the manufacturer’s instructions. RNA was isolated from 200 µL plasma using the miRNeasy Plasma/Serum Kit (Qiagen) according to the manufacturer’s instructions. The NanoDrop 1000 spectrophotometer (Thermo Fisher Scientific, Waltham, MA, USA) was used to quantify the RNA. cDNA was synthesized from either 1 µg (cell lines) or 0.1 µg (plasma) total RNA through random priming using the High-Capacity cDNA Reverse Transcription Kit (Applied Biosystems, Waltham, MA, USA) according to the manufacturer’s instructions. 

### 2.3. circRNA Enrichment Using RNase R

Degradation of linear RNA for circRNA enrichment by digestion using RNase R from Lucigen (LGC, Teddington, Middlesex, UK) was performed. Briefly, 2 µg RNA was digested using RNase R according to the protocol by Panda et al. [34] followed by immediate RNA isolation using the miRNeasy Kit (Qiagen) according to the manufacturer’s protocol. A control RNA reaction was also performed in parallel to the RNase R reaction but without Ribonuclease R. Both RNA with and without RNase R digestion were reverse transcribed to cDNA using High-Capacity cDNA Reverse Transcription Kit (Applied Biosystems) and RT-qPCR was performed for circRNA analysis. Data were analysed according to the protocol by Panda et al. [34] (Appendix A). 

### 2.4. circRNA Microarray

The isogenic LNCaP clones (*n* = 3) were analysed using the Arraystar Human circRNA Array version 2.0 (Arraystar, Rockville, MD, USA). The RNA sample preparation and microarray hybridization were done according to the manufacturer’s protocol as described previously in Greene et al. [29]. 

### 2.5. RT-qPCR and RT-ddPCR

RT-qPCR was performed using SYBR Green Master Mix (Applied Biosystems) on the 7500 Fast Real-Time PCR System (Applied Biosystems) using the following cycling conditions; hold: 50 °C for 20 s (1 cycle), hold: 95 °C for 10 min (1 cycle), denaturation: 95 °C for 15 s and annealing: 60 °C for 30 s (40 cycles). PCR reactions were prepared in a 96-well plate with a final volume of 20 µL per reaction: 10 µL 2X SYBR Green Master Mix, 1 µL forward (FWD) primer (10 µM), 1 µL reverse (REV) primer (10 µM), 7 µL molecular grade water and 1 µL cDNA. Data were normalised to GAPDH and fold change (FC) was calculated using the ∆∆ Ct relative quantification method. RT-ddPCR was performed using EvaGreen Supermix (Bio-Rad, Hercules, CA, USA) on the QX200™ Droplet Digital PCR System (Bio-Rad). Each PCR reaction was first prepared in a 1.5 mL tube (Eppendorf, Hamburg, Germany) with a final volume of 20 µL containing 10 µL 2X EvaGreen Supermix, 1 µL custom FWD primer (10 µM), 1 µL custom REV primer (10 µM) and 6 µL molecular grade water and 2 µL plasma cDNA or 7 µL molecular grade water and 1 µL cell lines cDNA. The 20 µL PCR reaction mix and 70 μL Droplet Generation Oil for EvaGreen (Bio-Rad) was subjected to a droplet generation procedure according to the manufacturer’s instructions. Following completion of droplet generation, 40 μL droplet contents was transferred into ddPCR™ 96-well plates (Bio-Rad), sealed with Pierceable Foil Heat Seal (Bio-Rad) using a PX1™ PCR Plate Sealer (Bio-Rad) and thermal cycled immediately using a T100™ Thermal Cycler (Bio-Rad) under the following conditions; hold: 95 °C for 5 min (1 cycle), denaturation: 95 °C for 30 s and annealing: 60 °C for 60 s (40 cycles) then, a signal stabilization step at 4 °C for 5 min, 90 °C for 5 min and lastly holding at 4 °C. A ramp rate of 2 °C/s was used at every cycling step to avoid droplet instability during rapid temperature alterations and to ensure each individual droplet attained uniform temperature during cycling. The QX200 Droplet Reader (Bio-Rad) was used to count individual droplets based on fluorescence intensity following completion of thermal cycling and analysed by QuantaSoft™ Analysis Pro software (Bio-Rad). Reactions were performed in triplicate and RT-ddPCR results were averaged as absolute copy number (copies/20 μL). The divergent primers overlapping the splice junction of hsa_circ_0001275 and hsa_circ_0001721 were custom designed using Circular RNA Interactome (https://circinteractome.nia.nih.gov/, accessed on 17 April 2018) [35] and synthesised by Integrated DNA Technologies (IDT) (Coralville, Iowa, USA). The convergent primers of GAPDH, Phospholipase C Like 2 (PLCL2), Keratin 1 (KRT1), B-cell leukaemia/lymphoma 11B (BCL11B), Ribosomal L1 Domain Containing 1 (RSL1D1) and Vacuolar Protein Sorting 72 Homolog (VPS72) were custom designed using the in silico PCR software suite in the UCSC Genome Browser (https://www.genome.ucsc.edu/cgi-bin/hgPcr, accessed on 17 April 2018) and synthesised by IDT. Primer sequences are given in Appendix A.

### 2.6. Sanger Sequencing 

The PCR product was sequenced by Source BioScience (Nottingham, UK). For PCR product preparation, 15 µL post-PCR reaction product was mixed with 6 µL ExoSAP-IT™ reagent (Applied Biosystems), incubated at 37 °C for 15 min to degrade remaining primers and nucleotides, followed by incubation at 80 °C for 15 min to inactivate ExoSAP-IT™ reagent. Following ExoSAP-IT™ treatment, the PCR products were prepared according to Source BioScience purified PCR product specifications.

### 2.7. Stable Overexpression of hsa_circ_0001275 and hsa_circ_0001721 

pCD25-ciR vector containing GFP marker (Geneseed Biotech, Guangzhou, China) (Appendix A) was used to overexpress circRNA of interest, hsa_circ_0001275 and hsa_circ_0001721. Primers to amplify both circRNAs were designed (Appendix A), and PCR products were sub-cloned into pCD25-ciR vector using the EcoRI and BamHI (Roche Diagnostics, Mannheim, Germany) restriction enzymes built into the primers for cloning purposes. LNCaP control cells were transfected with both an empty vector control (EVC) and the respective pCD25-ciR circRNA expressing plasmids using Lipofectamine^®^ 3000 (Life Technologies Corporation, Carlsbad, CA, USA) according to the manufacturer’s protocol to generate the following three cell sub-lines: (i) EVC, (ii) hsa_circ_0001275 (1275 Stable), (iii) hsa_circ_0001721 (1721 Stable). To select for stable integration of the individual plasmids, transfected cells were selected by treatment with geneticin (G418—Invivogen, San Diego, CA, USA) at a concentration of 450 µg/mL (defined as the optimal dose from a G418 killing curve, Appendix A) for more than 2 weeks. As the pCD25-ciR plasmid also expresses GFP (Appendix A), cells were monitored regularly for enrichment of GFP-expressing cells. At this point expanded stable clones were then sorted using a BD FACSMelody™ (BD Biosciences, San Jose, CA, USA) flow cytometer for GFP expression to create the relevant EVC and circRNA overexpressing clones used in the analyses presented. To establish that the relevant circRNAs had indeed been stably selected for circRNA overexpression was confirmed using RT-qPCR.

### 2.8. BrdU Proliferation Assay

Cells were seeded at 1 × 10³/well in a final volume of 100 µL/well in a 96-well plate and treated with enzalutamide (Stratech Scientific Ltd., Suffolk, UK) at a range of concentrations (0–80 µM) for a period of seven days with dimethyl sulfoxide (DMSO) serving as the vehicle control. Cell proliferation was measured using a Cell Proliferation ELISA, BrdU (Roche Diagnostics Ltd., Sussex, UK) according to the manufacturer’s protocol. Absorbance was measured on a Vesamax tuneable microplate reader (Molecular Devices, San Jose, CA, USA) at 450 nm with the reference wavelength set to 690 nm. The absorbance value of drug-treated cells was normalized to cells treated with DMSO which was set to 100%.

### 2.9. Patient Samples

Plasma samples were obtained from 44 patients with MCRPC collected as part of the Phase II trial of enzalutamide in combination with radium-223 (CTRIAL-IE 13-21, NCT02225704) [32]. Docetaxel chemotherapy-naive patients with MCRPC who progressed on ADT were enrolled between June 2015 to July 2017 and the trial closed in August 2021. The treatment regimen consisted of enzalutamide (160 mg per day, oral) in combination with 6 cycles of radium-223 (55 kBq/kg monthly, intravenous) followed by enzalutamide alone until disease progression, unacceptable toxicity or withdrawal of consent. A full description of the patient characteristics in this cohort has previously been described [32]. Blood specimens were collected at baseline before the start of treatment, at month 4 of treatment then every 3 months (±1 month) until disease progression or study withdrawal. Blood samples were collected in K2-EDTA collection tubes and centrifuged at 2000× *g*, 4 °C for 15 min within 60 min of collection for plasma isolation and stored at −80 °C. The clinical component of this study was nested within a trial registered with Cancer Trials Ireland (CTRIAL-IE 13-21) and with ClinicalTrials.gov (ClinicalTrials.gov identifier: NCT02225704) before the first patient was enrolled. This study received ethical approval by the Clinical Research Ethics Committee of the Cork Teaching Hospitals on 21 July 2014, REF: ECM51 D1/07/14. 

### 2.10. Statistical Analysis

The data were analysed using analysis of variance (ANOVA) and Student’s *t-*test using GraphPad Software 9.1.0 (San Diego, CA, USA) as appropriate. All results were summarised and presented as mean ± SEM followed by one-way/two-way ANOVA with Tukey’s or Friedman–Šídák multiple comparisons test as appropriate. A value of *p* < 0.05 was considered statistically significant. 

## 3. Results

### 3.1. hsa_circ_0001275 and hsa_circ_0000129 Are Up-Regulated in Enzalutamide Resistant Cells

The top 5 up-regulated circRNAs—as ranked by highest FC from our previous enzalutamide resistant cell line panel circRNAs microarray data [29]—along with their associated parental genes were validated via RT-qPCR (Table 1). 

Custom-designed divergent primers were used to amplify circRNAs across the backsplice junctions, instead of canonically spliced linear mRNA counterparts (Appendix A). In order to exclude the possibility of the linear form of RNA detection by the PCR primers, circRNA enrichment using RNase R for hsa_circ_0001275 was performed (Appendix A). Additionally, the hsa_circ_0001275 and hsa_circ_0001721 RT-qPCR product was subjected to Sanger sequencing and demonstrated a match to the backsplice junction sequences from the circinteractome database (Appendix A).

Out of the five circRNAs validated, two were in line with the microarray data. hsa_circ_0001275 was significantly up-regulated in Clone 1 vs. Control (FC 5.01 ± SEM 0.74, *p* = 0.006) (Figure 1a). hsa_circ_0001275 is an antisense circRNA, located on chromosome 3 and its associated parental gene PLCL2 was also significantly up-regulated in Clone 1 vs. Control and Clone 9 vs. Control (FC 7.23 ± SEM 0.06, *p* = 0.002 and FC 8.33 ± SEM 1.19, *p* = 0.001, respectively) (Figure 1b). The second validated circRNA was an exonic circRNA, located on chromosome 1—hsa_circ_0000129 was up-regulated in Clone 1 vs. Control (FC 2.56 ± SEM 0.21, *p* = 0.004), however, its associated parental gene VPS72 was not dysregulated between the cell lines (FC 0.79 ± SEM 0.02, *p* = 0.31) (Figure 1c,d, respectively). The parental gene for hsa_circ_0001721, CDK14, was of interest and selected for validation as our group had previously shown that the circRNA was up-regulated in the highly resistant clone compared to control [29]. CDK14 was also up-regulated in Clone 1 vs. Control (FC 3.51 ± SEM 0.64, *p* = 0.009) (Figure 1e).

There were no differences in the expression of hsa_circ_0033144 and hsa_circ_0026462 in Clone 1 vs. Control (Appendix A), whereas hsa_circ_0000673 was significantly down-regulated in Clone 1 vs. Control (FC 0.31 ± SEM 0.04, *p* = 0.022) (Appendix A) in contrast to the microarray result. The associated parental genes for hsa_circ_0033144, hsa_circ_0026462 and hsa_circ_0000673 were not differentially expressed in Clone 1 vs. Control (Appendix A).

### 3.2. hsa_circ_0001275 Overexpression Contributed to Enzalutamide Resistance

We further studied the effects of hsa_circ_0001275 and hsa_circ_0001721 overexpression on enzalutamide resistance. Both circRNAs were selected on the basis that their parental gene was also up-regulated in the strongly resistant clone compared to control. The transfection efficiency was initially determined by subjective estimation through examination under EVOS™ Digital Colour Fluorescence Microscope (Life Technologies Corporation) (Figure 2a–d) and confirmed quantitatively using RT-qPCR (Figure 3a,b). hsa_circ_0001275 was significantly overexpressed in 1275 Stable vs. Control and vs. EVC (*p* < 0.05) (Figure 3a). The 1275 Stable cell line demonstrated a significantly increased resistance to enzalutamide at every concentration tested (*p* < 0.05) except for 80 µM compared to EVC (Figure 4a). The 1275 Stable cell line showed significantly increased resistance to enzalutamide at every concentration tested (*p* < 0.05) except for 2 µM and 80 µM compared to the Control cell line (Appendix A). 

hsa_circ_0001721 was significantly overexpressed in 1721 Stable vs. Control and EVC (*p* < 0.05) (Figure 3b). The 1721 Stable cell line did not demonstrate enhanced resistance to enzalutamide compared to the EVC cell line (Figure 4b). The 1721 Stable cell line demonstrated significant resistance to enzalutamide at the 2, 5, 15 and 20 µM concentration range (*p* < 0.05) compared to the Control cell line (Appendix A).

### 3.3. circRNA Overexpression Did Not Significantly Change the Expression Level of Its Associated Parental Gene

The expression level of the parental gene that encodes its own specific circRNA was investigated in circRNA overexpression cell lines using RT-qPCR. There was no significant difference in the expression of PLCL2 in the cell lines that overexpressed hsa_circ_0001275 (*p* > 0.05, 1275 Stable vs. EVC and 1275 Stable vs. Control) (Figure 5a), nor CDK14 in 1721 Stable cell lines that overexpressed hsa_circ_0001721 (*p* > 0.05, 1721 Stable vs. EVC and 1721 Stable vs. Control) (Figure 5b).

### 3.4. Expression Pattern of hsa_circ_0001275 in Plasma of Patients with Prostate Cancer Treated with Enzalutamide 

As our data demonstrated a potential role of hsa_circ_0001275 overexpression with enzalutamide resistance, we further evaluated this circRNA as a biomarker of enzalutamide resistance clinically via RT-ddPCR detection. hsa_circ_0001275 expression was quantified in the plasma of 44 patients with MCRPC on the radium-223 in combination with enzalutamide in metastatic castration-resistant prostate cancer phase II trial (CTRIAL-IE 13-21, NCT02225704) where patients received 6 cycles of radium-223 in combination with enzalutamide followed by enzalutamide alone until disease progression or treatment intolerance. The plasma from four different time points was selected according to the PSA level. This consisted of baseline (prior to treatment, TP1), PSA nadir (absolute lowest PSA level on treatment, TP2), first PSA progression (an increase of ≥25% and absolute increase of ≥2 ng/mL of the PSA level from the nadir, TP3) and second PSA progression (PSA level higher than the first PSA progression, TP4). All 44 patients had plasma available for both TP1 and TP2 time points, whereas only 21 patients had plasma for all four time points since not all patients had PSA progression at the time of this study. Assay optimisation was initially performed using optimized primer concentration and a custom-designed gBlock (Appendix A) as positive control under a range of temperatures (Appendix A). This step showed that absolute quantification can precisely detect the expression of both circRNAs as shown by the clear separation of positive droplets from the negative droplet background (Appendix A). Furthermore, the expression of hsa_circ_0001275 was also detected in the isogenic LNCaP clones where both circRNAs were significantly up-regulated in the strongly enzalutamide resistant clone compared to control (*p* < 0.05) further confirming our initial RT-qPCR results (Appendix A). To estimate the accuracy between RT-ddPCR and RT-qPCR, we calculated the relative average copies/20 µL of transcript in Clone 1 to Control and Clone 9 to Control and compare the fold change in Clone 1 vs. Control and Clone 9 vs. Control for the two methodologies respectively. The results are presented in Appendix A. Overall, when compared across cell lines the methodologies have a similar pattern.

The expression of hsa_circ_0001275 in plasma samples showed a trend that mirrors the activity of the disease status according to the PSA level, however, this was not statistically significant (Figure 6a–d). All 44 patients had the first two time points available and their mean average copies/20 µL of hsa_circ_0001275 at TP2 (4 copies/20 µL) (plasma at time of PSA nadir) were lower than baseline TP1 (6 copies/20 µL) (prior to enzalutamide treatment) (Figure 6a). TP2 was the time point where the disease responded to enzalutamide treatment (Figure 6a) as shown by the mean PSA level which was significantly lower at TP2 (75 µg/L) compared to TP1 (10 µg/L) (*p* < 0.05) (Figure 6b). A subgroup analysis of 21 patients who had four available time points showed a similar trend where the mean average copies/20 µL of hsa_circ_0001275 was high at TP1 (9 copies/20 µL) then reduced at TP2 (6 copies/20 µL) but remained the same at TP3 (6 copies/20 µL) and increased again upon further PSA progression at TP4 (12 copies/20 µL) compared to TP3 (Figure 6c). The mean PSA at nadir, TP2 (20 µg/L) was significantly lower compared to TP1 (94 µg/L) and TP4 (186 µg/L) (*p* < 0.05), however, this was not significantly different compared to TP3 (90 µg/L) (Figure 6d). This indicated that the disease at TP3 is biochemically less resistant to enzalutamide compared to TP4.

## 4. Discussion

circRNAs were first discovered in the early 1990s [20], however, they were initially believed to represent by-products of splicing error until 2012 when a seminal paper described the identification of circRNAs in RNA-seq data generated from samples of paediatric acute lymphoblastic leukaemia suggesting that they may be more abundant and of more significant importance than previously perceived [41]. Recently, we and others showed that circRNAs may play important roles in numerous malignancies including prostate cancer [28,29]. However, the molecular function of circRNAs in drug-resistant prostate cancer and their potential as biomarkers to inform treatment strategies remain elusive. A previous study by Wu et al. demonstrated that hsa_circ_0001427 downregulation contributed to enzalutamide resistance, while Cao et al. showed that a circRNA derived from the AR was overexpressed when prostate cancer progressed to CRPC and could be detected in plasma [28,42]. Furthermore, our group identified a number of circRNAs that were differentially expressed in a model of enzalutamide resistance further supporting the notion that circRNAs play a role in enzalutamide resistance and potentially have utility as biomarkers to monitor for the emergence of enzalutamide resistance and prostate cancer progression [29]. 

From our initial array results, we examined the top 5 circRNAs ranked by highest FC in the strongly resistant cell line (*p* < 0.05, Clone 1 vs. Control) from our published circRNA microarray data [29] and validated them along with their associated parental gene via RT-qPCR. We found that hsa_circ_0001275 and hsa_circ_0000129 were up-regulated in the strongly resistant cell lines compared to Control (*p* < 0.05) validating our previous microarray results. Moreover, when examined, the associated parental gene PLCL2 from which hsa_circ_0001275 is derived, was also significantly up-regulated (*p* < 0.05, Clone 1 vs. Control) (Figure 1b). Interestingly, a previous study identified PLCL2 as part of a 23-gene expression panel that predicts metastatic lethal prostate cancer outcomes in patients with localised disease treated surgically [36] thus, further strengthening the possible association of this circRNA with prostate cancer. In contrast, VPS72 from which hsa_circ_0000129 is derived showed no significant difference in expression (Figure 1c).

Numerous studies have demonstrated that circRNAs are differentially expressed in various human diseases [29,41,43]. Despite this, there is a paucity of literature that has investigated circRNAs’ molecular function [44] and even fewer have investigated the role of circRNA in drug resistance [28]. In order to study the effects of circRNA on enzalutamide resistance, circRNAs of interest were overexpressed in the Control cell line that was sensitive to enzalutamide. hsa_circ_0001275 and hsa_circ_0001721 were selected for overexpression experiments instead of hsa_circ_0000129 since both associated parental genes were significantly up-regulated (*p* < 0.05, Clone 1 vs. Control) and known to be associated with invasion and metastasis (a known hallmark in cancer) [36,45], and hsa_circ_0001721 was previously validated by our group as being upregulated and associated with resistance to enzalutamide [29]. When stably overexpressed, hsa_circ_0001275 was shown to be associated with increased resistance to enzalutamide (*p* < 0.05) as determined using proliferation assays (Figure 4a). In contrast, the same proliferative effect was not demonstrated in the hsa_circ_0001721 overexpressing stable cell line suggesting that increased hsa_circ_0001721 does not contribute to enzalutamide resistance (Figure 4b). This indicates only certain up-regulated circRNAs may play a role in enzalutamide resistance by acting as a key regulator through competitive endogenous RNA (ceRNA) networks. Studies have shown a historical role of miRNA “sponging” in circRNA-miRNA–mRNA networks as playing important roles in both gene regulation and carcinogenesis [25]. For example, Xu et al. recently showed that dexamethasone-induced osteoblast growth inhibition can be reversed by downregulating hsa_circ_0001275 which mediated the miR-377/CDKN1B axis in an effort to discover new treatment for osteoporosis [46]. Furthermore, Formosa et al. previously demonstrated a role for miR-377 as a tumour suppressor miRNA targeting FZD4, a gene involved in epithelial-to-mesenchymal transition in prostate cancer [47]. The top five most likely miRNA binding sites for hsa_circ_0001275 were predicted using Arraystar’s miRNA prediction software, and four out of the five predicted miRNAs were hypothesized as tumour suppressor miRNAs in the literature (Appendix A). Thus, hsa_circ_0001275 may contribute to the development of enzalutamide resistance through these ceRNA networks, however, further investigations are required to fully delineate the circRNA-miRNA-mRNA network. The variability of circRNA expression is however, not dependent on its parental gene expression as shown by Vo et al. [48] and vice versa as shown in our data where expression of the circRNA parental gene was not significantly altered when its circRNA was overexpressed (Figure 5a,b). Thus far, studies in the literature have demonstrated that circRNAs are universally downregulated in proliferative cells across different cancer subtypes, which may be attributed to circRNA concentration dilution upon fast cell division as described by Bachmayr-Heyda et al. [26] and further supported by the findings from Gruner et al. [49] showing accumulation of circRNAs in non-proliferative cells such as ageing nervous tissue. Furthermore, Vo et al. [48] further demonstrated that by using a potent kinase inhibitor to slow down cell growth by decreasing cellular proliferation, the global abundance of circRNAs in LNCaP became elevated and this abundance is independent of the parental gene associated with the circRNA. Our data so far, showed that hsa_circ_0001275 was upregulated along with its parental gene PLCL2 in Clone 1. PLCL2 is associated with aggressive prostate cancer [36]. In the hsa_circ_0001275 (circ1275 Stable) overexpressed cell line, which was more resistant to enzalutamide, there was no difference in the expression of PLCL2 compared to control. PLCL2, therefore, does not contribute to the increase in resistance. As such, from the literature mentioned above, we believe that the enzalutamide resistance seen in our data was not due to the increase in malignancy/proliferation in general, however, the precise role of this circRNA on the enzalutamide mechanism of action is under investigation at this point in time.

Interest in circRNAs as candidate biomarkers in oncology has been established by their known dysregulation across different types of malignancy [50] and their ability to be detected in human bodily fluids such as saliva, blood and gastric fluid [51,52,53]. In order to further investigate the value of circRNA as a biomarker to inform drug resistance, we, therefore, investigated the level of hsa_circ_0001275 in plasma samples of clinical trial (CTRIAL-IE 13-21, NCT02225704) patients with prostate cancer on enzalutamide treatment via RT-ddPCR. These results suggested a trend that corresponded to the activity of the disease defined by the PSA level of patients on enzalutamide treatment, however, it did not reach statistical significance likely due to small cohorts and/or copy numbers were close to the limit of detection for RT-ddPCR. Previous studies have reported that circRNAs have tissue specificity [54] and thus may explain the low abundance of hsa_circ_0001275 in plasma especially if it is not actively secreted by tumour tissue. We did not examine whether or not the particular circRNAs studied were secreted from prostate cancer cells. Our data suggests that hsa_circ_0001275 was not expressed abundantly in plasma and currently limits its potential use as a liquid-based predictive biomarker of enzalutamide resistance. It may be that in the future increased sensitivity may be obtained using a panel of circRNAs.

There are some limitations in our study that should be acknowledged. One of the limitations of the current study is that the study used a single isogenic cell line model of enzalutamide resistance developed by Korpal et al. [33]. In the future, we aim to explore the overexpression of this circRNA in additional enzalutamide-sensitive prostate cancer cell lines to further delineate its association with drug resistance. An additional limitation is that the cohort of patients in the study was relatively small. After initial statistical analysis, we found that our circRNA data were positively skewed. Therefore, we performed the analysis using the Wilcoxon signed-rank test for patients with two time points (Figure 6a,b) and the Friedman test for those with four time points (Figure 6c,d). It has been argued that a sample size of more than five in each group is sufficient, however, the expected size of the difference across groups is also important. Our patient sample size was small which made it difficult to ascertain the statistical significance in particular to confidence interval and *p*-values. Additionally, despite utilising RT-ddPCR, low copy numbers of hsa_circ_0001275 were present in plasma samples, which may have contributed to large differences across groups. It could be argued that a pre-amplification step such as that used by Karousi et al. [55], may have helped in the detection of circRNAs, however, it must be noted that this approach can cause issues such as multiple PCR products for the same circRNA caused by rolling cDNA from the circRNA as exemplified in the study by Chen et al. [56]. Moving forward for such studies, it may be important to first evaluate their expression using an explant or organoid based protocol to determine whether or not individual circRNAs of interest can be detected within the conditioned media, which will pave the way for more robust studies to test for their presence in patient plasma. Second, we performed a time-point analysis in this study according to PSA levels only (Figure 6). PSA level is not the only surrogate maker to inform disease progression, however, increasing PSA is usually the initial indication of tumour regrowth which is then succeeded by progression of disease on imaging and ultimately symptomatic clinical presentations [57]. Thus, the time points selected may not be a solid reflection of disease progression and tumour burden. Third, this study involved combination treatment of six cycles of alpharadium and enzalutamide followed by enzalutamide maintenance until disease progression or treatment intolerance and the cohort was relatively small. Ideally, the results presented in this study should be examined in a larger study with patients on enzalutamide alone. Fourth, we did not study the level of circRNA expression in tissue samples as they were not available since tissue biopsy upon disease progression were not performed routinely as part of the trial.

## 5. Conclusions

We found hsa_circ_0001275 up-regulated in enzalutamide-resistant cell lines with data suggesting a protective role against enzalutamide in vitro. While our current research suggests that hsa_circ_0001275 plays a role in resistance to enzalutamide, the functional mechanisms underpinning this have yet to be elucidated for example effects on tumourigenicity or migration and these will form the basis of future studies. Our data also showed that hsa_circ_0001275 is not expressed abundantly in plasma. Although the trend of expression mirrors disease activity according to the PSA level suggest its association with enzalutamide resistance, further studies are required. The treatment landscape of prostate cancer has evolved rapidly in the last decade due to the greater availability and accessibility of therapeutic agents. Despite this arsenal of treatments, patients eventually progress due to drug resistance presenting a difficult conundrum. These novel circRNAs have potential in combating drug resistance and it seems certain that many that are involved remain undescribed. We believe this study provides beneficial groundwork for further exploration of circRNAs potential as biomarkers and therapeutic targets for drug-resistant prostate cancer.

## Figures and Tables

**Figure 1 cancers-13-06383-f001:**
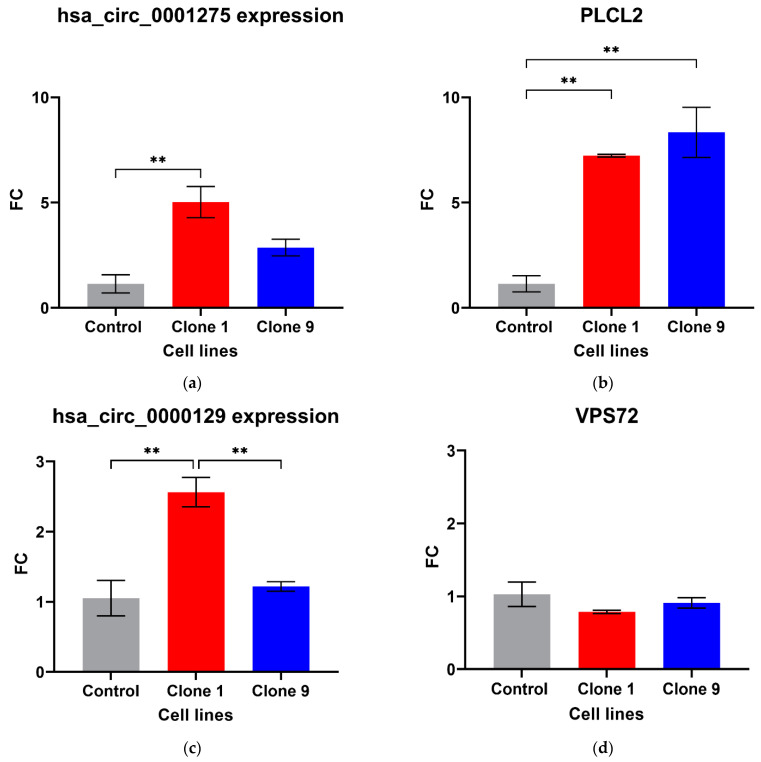
Validation of circRNAs and associated parental genes in the enzalutamide panel (**a**) hsa_circ_0001275, (**b**) PLCL2 parental gene, (**c**) hsa_circ_0000129, (**d**) VPS72 parental gene, (**e**) CDK14 parental gene for hsa_circ_0001721 (the circRNA expression has been previously presented [29]). Data graphed as the mean ± SEM (*n* = 3). Data were analysed using an ordinary one-way ANOVA followed by a Tukey’s post hoc test. (** *p* ≤ 0.01).

**Figure 2 cancers-13-06383-f002:**
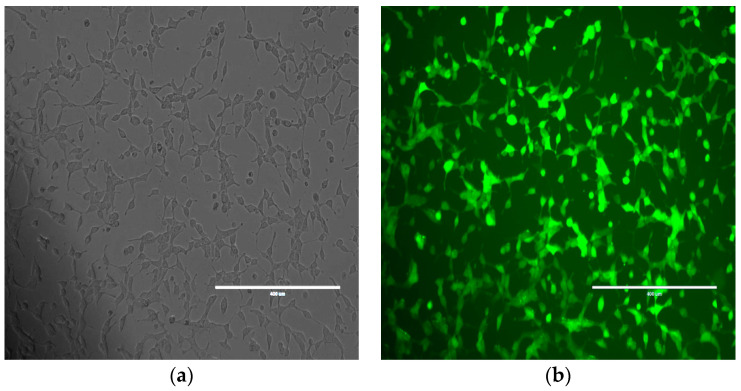
Representative images of Control cell line transfected with hsa_circ_0001275 using (**a**) bright field and (**b**) fluorescence microscopy and hsa_circ_0001721 using (**c**) bright field and (**d**) fluorescence microscopy. All scale bars are 400 µm. All images are 10× magnification.

**Figure 3 cancers-13-06383-f003:**
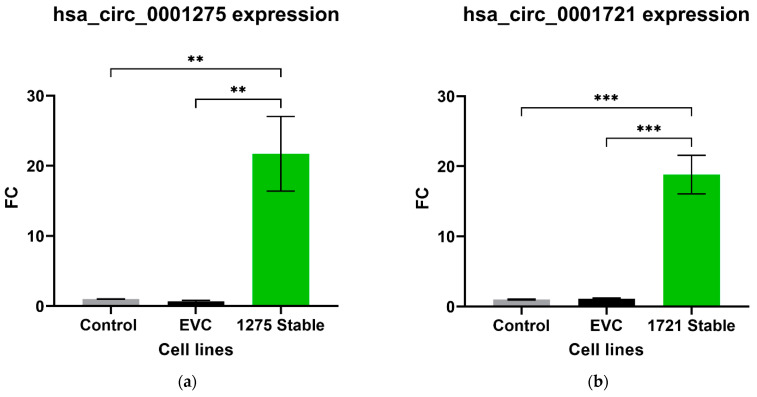
Relative expression of (**a**) hsa_circ_0001275 and (**b**) hsa_circ_0001721 in the stably transfected cell line compared to Control and EVC cell lines. Data graphed as the mean ± SEM (*n* = 3). Data were analysed using an ordinary one-way ANOVA followed by a Tukey’s post hoc test. (** *p* ≤ 0.01, *** *p* ≤ 0.001).

**Figure 4 cancers-13-06383-f004:**
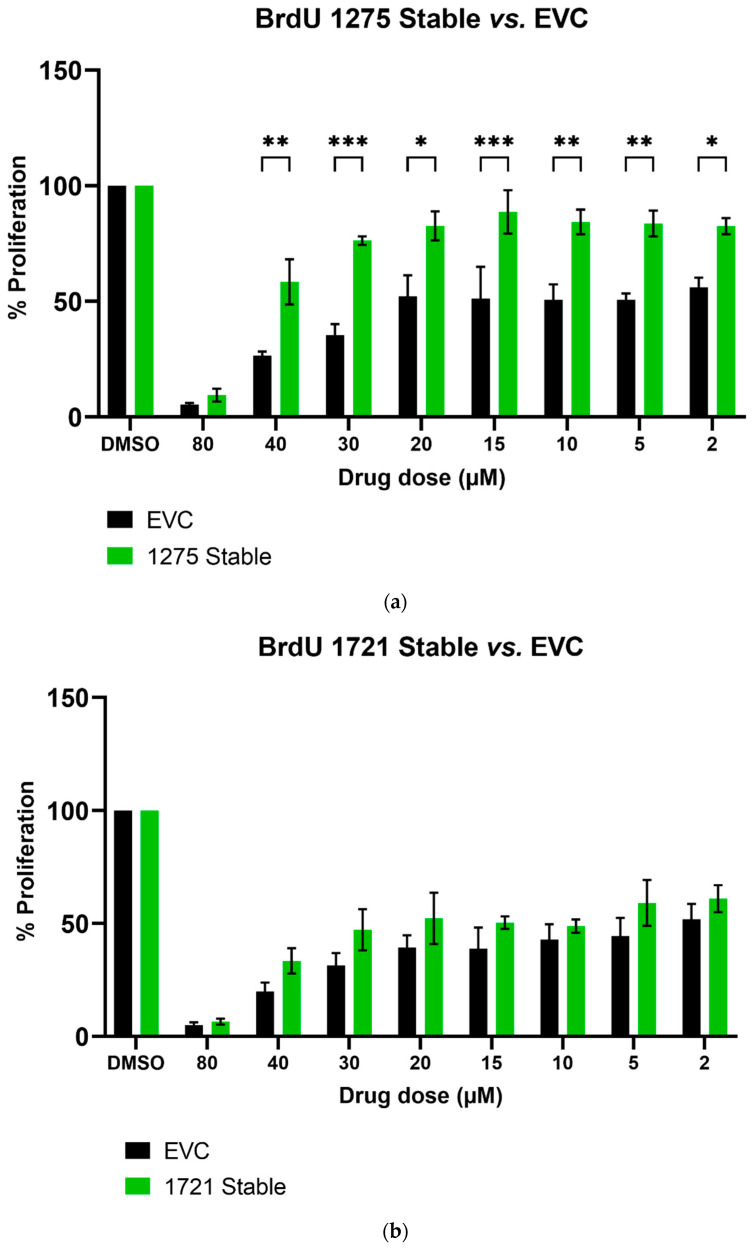
The effect of enzalutamide on the proliferative rate of EVC and cell line overexpressing (**a**) hsa_circ_0001275 (1275 Stable) and (**b**) hsa_circ_0001721 (1721 Stable). Proliferation was measured using BrdU ELISA. Data graphed as the mean ± SEM (*n* = 3). Analysis was performed using a two-way ANOVA followed by a Šídák’s post hoc test. (* *p* ≤ 0.05, ** *p* ≤ 0.01, *** *p* ≤ 0.001). Dimethyl sulfoxide (DMSO).

**Figure 5 cancers-13-06383-f005:**
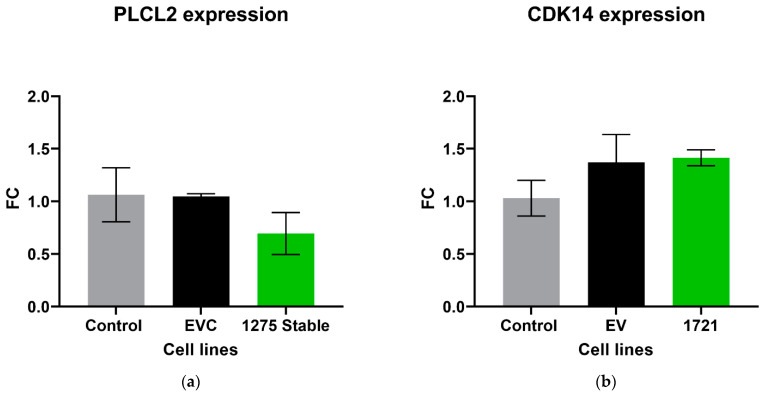
Relative expression of (**a**) PLCL2 in the stably transfected cell line which overexpressed hsa_circ_0001275 (1275 Stable) and (**b**) CDK14 in the stably transfected cell line which overexpressed hsa_circ_0001721 (1721 Stable). Data graphed as the mean ± SEM (*n* = 3). Data were analysed using an ordinary one-way ANOVA followed by a Tukey’s post hoc test.

**Figure 6 cancers-13-06383-f006:**
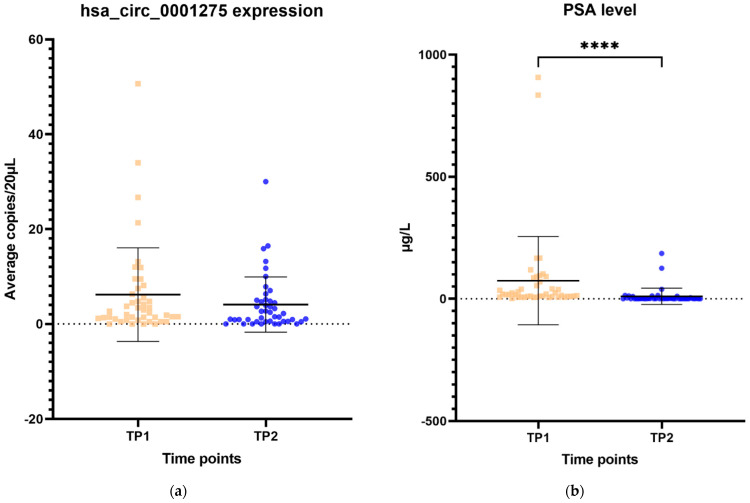
Scatter plot of hsa_circ_0001275 level in plasma samples and PSA level. (**a**) hsa_circ_0001275 level in plasma samples and (**b**) PSA level of all 44 patients with first 2 available time points. Data graphed as mean ± SD (*n* = 44) and analysed using the Wilcoxon signed-rank test. (**c**) hsa_circ_0001275 level in plasma samples and (**d**) PSA level of 21 patients with all 4 time points available. Data graphed as the mean ± SD (*n* = 21) and analysed using one-way ANOVA with Friedman post hoc test. (* *p* ≤ 0.05, **** *p* ≤ 0.0001). Plasma at TP1 was selected as baseline prior to enzalutamide treatment, plasma at TP2 was selected during PSA nadir, plasma at TP3 was selected when PSA first progressed and plasma at TP4 was selected on second PSA progression.

**Table 1 cancers-13-06383-t001:** Top five up-regulated circRNAs in Clone 1 vs. Control based on FC with their associated parental gene (with hypothesised gene function) adapted from Greene et al. [29].

circRNA	FC/(*p*-Value)	Parental Gene	Parental Gene Function
hsa_circ_0001275	5.8 (*p* = 0.047)	PLCL2	Part of a 23-gene signature, which predicted metastatic lethal prostate cancer outcomes [36].
hsa_circ_0026462	5.7 (*p* = 0.026)	KRT1	Target receptor overexpressed in breast cancer cells [37].
hsa_circ_0033144	5.2 (*p* = 0.012)	BCL11B	Methylation occurs in prostate cancer [38].
hsa_circ_0000673	4.2 (*p* = 0.038)	RSL1D1	Associated with an aggressive phenotype and a poor prognosis in patients with prostate cancer [39].
hsa_circ_0000129	3.9 (*p* = 0.039)	VPS72	May have a role in regulating haematopoietic stem cell activity [40].

## Data Availability

The microarray data discussed in this publication have been deposited in NCBI’s Gene Expression Omnibus [58] and are accessible through GEO Series accession number GSE118959 (https://www.ncbi.nlm.nih.gov/geo/query/acc.cgi?acc=GSE118959, accessed on 31 October 2018).

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
