# Peer review of "hsa_circ_0001275 Is One of a Number of circRNAs Dysregulated in Enzalutamide Resistant Prostate Cancer and Confers Enzalutamide Resistance In Vitro"

_cancers, 2021, doi:10.3390/cancers13246383_

Round 1

Reviewer 1 Report

The Comments have well responded to my comments.

Reviewer 2 Report

I believe in this form the paper may be published. However, the title is still can be improved.

This manuscript is a resubmission of an earlier submission. The following is a list of the peer review reports and author responses from that submission.

Round 1

Reviewer 1 Report

This work is an important step in the search for new biomarkers of malignant disease progression. circular RNA as biomarkers suggest several critical advantages compared to other biological molecules that can be detected in biological fluids. Authors well describe the design of the study. Important part of the study is a validation of the process using in vitro system. generated prostate cancer cell lines well present clinical stages of prostate cancer. The clinical part of the study need an modifications t , but authors clearly understand this and present a plan for  improvement of the study.

For my opinion the title of the manuscript is confusing and need to be edited. Material and methods section  may be clarified. In general this is very useful work for good future potentials.

Author Response

This work is an important step in the search for new biomarkers of malignant disease progression. circular RNA as biomarkers suggest several critical advantages compared to other biological molecules that can be detected in biological fluids. Authors well describe the design of the study. Important part of the study is a validation of the process using in vitro system. generated prostate cancer cell lines well present clinical stages of prostate cancer. The clinical part of the study need an modifications t , but authors clearly understand this and present a plan for  improvement of the study.

For my opinion the title of the manuscript is confusing and need to be edited. Material and methods section  may be clarified. In general this is very useful work for good future potentials.

.>>We thank the reviewer for raising these suggestions and comments.

We have edited the title of the manuscript.

The amended title is as follows:

“hsa_circ_0001275 is one of a number of circRNAs dysregulated in enzalutamide resistant prostate cancer and confers enzalutamide resistance in vitro.”

We have now added new text to the material and methods section.

-The new text has been added within the manuscript page 3, line 108 to 113.

The amended text is as follows:

“The STR profiles for the LNCaP Control, C1 and C9 cells were compared against the ATCC STR database (https://www.atcc.org/search-str-database ). The matching criterion is based on an algorithm that compares the number of shared alleles between two cell line samples, expressed as a percentage. Cell lines with ≥80% match are considered to be related; derived from a common ancestry. Using the following STR criteria: (AMEL) + the following loci: D5S818, D13S317, D7S820, D16S539, vWA, TH01, TPOX, CSF1PO. The results for identity using the ATCC algorithm were as follows: LNCAP Control: 94% (Pass), LNCaP C1: 94% (Pass), LNCaP C9: 94% (Pass).”

-The new text has been added within the manuscript page 3, line 117 to 119.

The amended text is as follows:

“RNA was isolated from 200 µL plasma using the miRNeasy Plasma/Serum Kit (Qiagen) according to manufacturer’s instructions. The NanoDrop 1000 spectrophotometer (Thermo Fisher Scientific, MA, USA) was used to quantify the RNA.”

-The new text has been added within the manuscript page 4, line 141 to 144.

The amended text is as follows:

“Each PCR reaction was first prepared in 1.5 mL tube (Eppendorf, Hamburg, Germany) with a final volume of 20 µL containing 10 µL 2X EvaGreen Supermix, 1 µL custom FWD primer (10 µM), 1 µL custom REV primer (10 µM) and 6 µL molecular grade water and 2 µL plasma cDNA or 7 µL molecular grade water and 1 µL cell lines cDNA.”

-The new text has been added within the manuscript page 5, line 196 to 197.

The amended text is as follows:

“A full description of the patient characteristics in this cohort has previously been described  [32].”

Reviewer 2 Report

In this research article, the authors aimed at correlating circRNA expression with enzalutamide resistance in prostate cancer. The manuscript is sound and interesting; however, I have the following concerns:

  • Introduction, lines 94-95: The authors mention that there is a scarcity of literature describing the role of circRNAs in enzalutamide resistance. However, as there are such instances in current literature, these should be mentioned. Papatsirou et al. have recently summarized these cases (Per Med. 2020 Nov;17(6):469-490. doi: 10.2217/pme-2020-0103).
  • As the authors have applied RT-qPCR and RT-ddPCR, it would be interesting to compare these two methods regarding their accuracy in circRNA quantification. Moreover, the authors could compare their qPCR assay with other assays in literature, some of which also included circRNA pre-amplification (Karousi et al., Int. J. Mol. Sci. 2020, 21, 8867. https://doi.org/10.3390/ijms2122886)
  • A table presenting the patient characteristics should be added.
  • The authors have created clones stably expressing circRNAs. As this is a major change for the cells, the authors could discuss how this modification has affected the signaling cascades implicated in prostate cancer progression and therapy resistance, as circRNAs have proved to be master regulators of signaling pathways involved in cancer progression (Cancers. 2021; 13(11):2744. https://doi.org/10.3390/cancers13112744).
  • The authors need to discuss the small sample size used in biostatistics and be modest regarding their conclusions drawn from these results.

Author Response

Introduction, lines 94-95: The authors mention that there is a scarcity of literature describing the role of circRNAs in enzalutamide resistance. However, as there are such instances in current literature, these should be mentioned. Papatsirou et al. have recently summarized these cases (Per Med. 2020 Nov;17(6):469-490. doi: 10.2217/pme-2020-0103).

We thank the reviewer for this comment and apologise for the oversight on our behalf. We have now added new text to the introduction section. The new text has been added within the manuscript page 3, line 84 to 89.

The amended text is as follows:

“A number of circRNA biomarkers have been investigated in prostate cancer as summarised by Papatsirou et al. [27], however, there are few studies in the literature describing the role of circRNA in enzalutamide resistance in vitro [28,29]. For instance, Wu et al. demonstrated that the downregulation of circRNA17 resulted in the overexpression of AR-V7 leading to enzalutamide resistance in prostate cancer cell lines [28] and Greene et al. found that the downregulation of hsa_circ_0004870 may lead to enzalutamide resistance through the regulation of AR-V7 [29].”

As the authors have applied RT-qPCR and RT-ddPCR, it would be interesting to compare these two methods regarding their accuracy in circRNA quantification.

We thank the reviewer for raising this suggestion. We have  compared the two methods as regards accuracy, and the results have been incorporated into the manuscript as Supplementary Table S4. In addition, we have added text to the manuscript page 12 and 13, line 317 to 320.

The amended text is as follows:

“To estimate the accuracy between RT-ddPCR and RT-qPCR, we calculated the relative average copies/20µL of transcript in Clone 1 to Control and Clone 9 to Control and compare the fold change in Clone 1 vs. Control and Clone 9 vs. Control for the two methodologies respectively. The results are presented in Supplementary Table S4. Overall, when compared across cell lines the methodologies have a similar pattern.”

“Moreover, the authors could compare their qPCR assay with other assays in literature, some of which also included circRNA pre-amplification (Karousi et al., Int. J. Mol. Sci. 2020, 21, 8867. https://doi.org/10.3390/ijms2122886)”

This is a very interesting point. Whilst pre-amplification steps have been used in the literature, we feel that this represents a problem as exemplified by the study of Chen and colleagues (https://doi.org/10.1080/13102818.2017.1398596), found that extended RT incubation time resulted in multiple rolling cDNA PCR products from circRNA, which interfered with the qPCR analysis. Therefore, we did not include a pre-amplification step in our analysis for this reason. We have now added a sentence to the discussion in the manuscript page 16 and 17, line 428 to 431.

The amended text is as follows:

“It could be argued that a pre-amplification step such as that used by Karousi et al. [55], may help in the detection of circRNAs, it must be noted that this approach can cause issues such as multiple PCR products for the same circRNA caused by rolling cDNA from the circRNA as exemplified in the study by Chen et al. [56].”

A table presenting the patient characteristics should be added.

We thank the reviewer for this suggestion. A description of the patient characteristics was published as part of the clinical study (DOI: 10.1177/17588359211042691). We have added a sentence to this effect in the methods section under patient samples in the manuscript page 5, line 196 to 197:

The amended text is as follows:

“A full description of the patient characteristics in this cohort has previously been described [32].”

The authors have created clones stably expressing circRNAs. As this is a major change for the cells, the authors could discuss how this modification has affected the signaling cascades implicated in prostate cancer progression and therapy resistance, as circRNAs have proved to be master regulators of signaling pathways involved in cancer progression (Cancers. 2021; 13(11):2744. https://doi.org/10.3390/cancers13112744).  

Again, we thank the reviewer for this suggestion to discuss the possibility of how overexpression of hsa_circ_0001275 might affect the signalling cascade implicated in therapy resistance. We have now added the circRNA predicted target miRNA with hypothesized associated target mRNA for hsa_circ_0001275 suggesting the “sponging” mechanism in the circRNA-miRNA-mRNA network to the discussion in the manuscript page 15 and 16, line 379 to 389. Additionally a table of the hsa_circ_0001275 predicted miRNA with its function and hypothesized associated target gene and function have been incorporated into the manuscript as supplementary Table S5.

The amended text is as follows:

“Studies have shown a historical role of miRNA “sponging” in circRNA-miRNA-mRNA networks as playing important roles in both gene regulation and carcinogenesis [25]. For example, Xu et al. recently showed that dexamethasone-induced osteoblast growth inhibition can be reversed by downregulating hsa_circ_0001275 which mediated the miR-377/CDKN1B axis in an effort to discover new treatment for osteoporosis [46]. Furthermore, Formosa et al. previously demonstrated a role for miR-377 as a tumor suppressor miRNA targeting FZD4, a gene involved in epithelial-to-mesenchymal transition in prostate cancer [47]. The top five most likely miRNA binding sites for hsa_circ_0001275 were predicted using Arraystar’s miRNA prediction software, and four out of the five predicted miRNAs were hypothesized as tumor suppressor miRNAs in the literature (Supplementary Table S5). Thus, hsa_circ_0001275 may contribute to development of enzalutamide resistance through these ceRNA networks, however further investigations are required to fully delineate the circRNA-miRNA-mRNA network.”

The authors need to discuss the small sample size used in biostatistics and be modest regarding their conclusions drawn from these results.

We agree that the sample size used is small and have updated the discussion sections in the manuscript page 16 and 17, line 421 to 431.

The amended text is as follows:

 “An additional limitation is that the cohort of patients in the study is relatively small. After initial statistical analysis, our circRNA data was positively skewed. Therefore, we performed the analysis using the Wilcoxon signed rank test for patients with two time points (Figure 6a and 6b) and the Friedman test for those with four time points (Figure 6c and 6d). It has been argued that a sample size of more than five in each group is sufficient, however, the expected size of the difference across groups is also important. Our patient sample size was small which made it difficult to ascertain the statistical significance in particular to confidence interval and p-values. Additionally, despite utilising RT-ddPCR, low copy numbers of hsa_circ_0001275 were present in plasma samples, which may have contributed to the large difference across groups. It could be argued that a pre-amplification step such as that used by Karousi et al. [55], may have helped in the detection of circRNAs, however, it must be noted that this can cause issues such as multiple PCR products for the same circRNA caused by rolling cDNA from the circRNA as exemplified in the study by Chen et al. [56].”

We have also updated the conclusion section drawing a more modest conclusion from the results in the manuscript page 17, line 445 to 455.

The amended text is as follows:

“We found hsa_circ_0001275 up-regulated in enzalutamide resistant cell lines with data suggesting a protective role against enzalutamide in vitro. While our current research suggests that hsa_circ_0001275 plays a role in resistance to enzalutamide, the functional mechanisms underpinning this have yet to be elucidated for example effects on tumourigenicity or migration and this forms the basis of future studies. Our data also showed that hsa_circ_0001275 is not expressed abundantly in plasma. Though the trend of expression mirrors disease activity according to PSA level suggest its association with enzalutamide resistance, further studies are required. The treatment landscape of prostate cancer has evolved rapidly in the last decade due to greater availability and accessibility of therapeutic agents. Despite this arsenal of treatments, patients eventually progress due to drug resistance presenting a difficult conundrum. These novel circRNAs have potential in combating drug resistance and it seems certain that many that are involved remain undescribed. We believe this study provides beneficial ground work for further exploration of circRNAs potential as biomarkers and therapeutic targets for drug resistant prostate cancer.”

Reviewer 3 Report

Dr.Lim et al. studied by qRT-PCR the expression level of selected circRNAs based on previous work on microarray analysis in one clone derived from parental LNCAP and made resistant to enzalutamide .

Hsa_circ_0001275 was found up-regulated in clone 1 and its overexpression in parental LnCAP increase proliferation under the treatment with enzalutamide.  Even if low levels of this circRNA were found in plasma from patients treated with enzalutamide a trend toward a correlation with PSA level was observed.

which is the proliferation rate of 1275 in comparison with LNCAP cells in the absence of drug?

This is an interesting preliminary work on functional analysis of Hsa_circ_0001275  in LNCAP, however these results are not conclusive and I suggest to perform additional experiments of overexpression.

Please try to overexpress this circRNA in at least other two enzalutamide sensitive prostate cancer cells to induce enzalutamide resistance.

what about overexpression of this circRNA and resistance to other drug?

What about overexpression of this circRNA and tumourigenicity or invasive growth?

How we can attribute the role of this circRNA on enzalutamide mechanism of action and not to increase malignancy in general?

Could you provide other models of enzalutamide resistance and find this marker again?

Minor revision: Please add STR profile of clone 1, 9 and parental LNCAP

Please add starting plasma volume and the RNA quantification method applied

Author Response

which is the proliferation rate of 1275 in comparison with LNCAP cells in the absence of drug?

We have analysed the BrdU data to evaluate the proliferation rate of circRNA over expressed cell lines in comparison to LNCaP cells in the absence of drug as requested. 

There were no significant differences in the proliferation rate between 1275 Stable vs. Control (107% vs. 100%) and EVC vs. Control (99% vs. 100%) in the absence of enzalutamide treatment (untreated) (Figure 1a).

There were no significant differences in the proliferation rate between 1721 Stable vs. Control (110% vs. 100%) and EVC vs. Control (102% vs. 100%) in the absence of enzalutamide treatment (untreated) (Figure 1b).                                              

Figure 1. The proliferative rate of Control, EVC and cell line overexpressing (a) hsa_circ_0001275 (1275 Stable) and (b) hsa_circ_0001721 (1721 Stable) in the absence of enzalutamide. Proliferation was measured using BrdU ELISA. Data graphed as mean ± SEM (n=3). Data analysed using an ordinary one-way ANOVA followed by a Tukey’s post-hoc test.

This is an interesting preliminary work on functional analysis of Hsa_circ_0001275  in LNCAP, however these results are not conclusive and I suggest to perform additional experiments of overexpression.

  • Please try to overexpress this circRNA in at least other two enzalutamide sensitive prostate cancer cells to induce enzalutamide resistance.

We thank the reviewer for raising this interesting point. However, this is beyond the scope of the current paper as the focus was to explore circRNA expression in an isogenic cell line panel of enzalutamide resistance in LNCaP cells (PMID: 23842682). We have expanded the limitation section of the discussion to include this in the manuscript page 16, line 418 to 421.

The amended text is as follows:

“One of the limitations of the current study is that the study used a single isogenic cell line model of enzalutamide resistance developed by Korpal et al. [33]. In the future, we aim to explore the overexpression of this circRNA in additional enzalutamide sensitive prostate cancer cell lines to further delineate its association with drug resistance.” 

  • what about overexpression of this circRNA and resistance to other drug?

This is a very interesting suggestion, however, it is outside the scope of the current study,  which was designed to evaluate circRNAs in enzalutamide resistance. It is something that will be taken on board for future studies and we thank the reviewer for this suggestion.

  • What about overexpression of this circRNA and tumourigenicity or invasive growth?

We have now added this as a statement in the conclusion in the manuscript page 17, line 446 to 448.

The amended text is as follows:

“While our current research suggests that hsa_circ_0001275 plays a role in resistance to enzalutamide, the functional mechanisms underpinning this have yet to be elucidated, for example effects on tumourigenicity or migration and these will form the basis of future studies.”

  • How we can attribute the role of this circRNA on enzalutamide mechanism of action and not to increase malignancy in general?

We thank the reviewer for raising this point. We have now added the points below to the discussion in the manuscript page 16, line 391 to 403.

The amended text is as follows:

“Thus far, studies in the literature have demonstrated that circRNAs are universally downregulated in proliferative cells across different cancer subtypes, which may be attributed to circRNA concentration dilution upon fast cell division as described by Bachmayr-Heyda et al. [26] and further supported by the findings from Gruner et al. [49] showing accumulation of circRNAs in non-proliferative cells such as ageing nervous tissue. Besides, Vo et al. [48] further demonstrated that by using a potent kinase inhibitor to slow down cell growth by decreasing cellular proliferation, the global abundance of circRNAs in LNCaP became elevated and this abundance is independent of the parental gene associated with the circRNA. Our data so far, showed that hsa_circ_0001275 was upregulated along with its parental gene PLCL2 in Clone 1. PLCL2 is associated with aggressive prostate cancer [36]. In the hsa_circ_0001275 (circ1275 Stable) overexpressed cell line, which was more resistant to enzalutamide, there was no difference in the expression of PLCL2 compared to control. PLCL2 therefore does not contribute to the increase in resistant. As such, from the literature mentioned above, we believe that the enzalutamide resistance seen in our data were not due to increase malignancy/proliferation in general, however the precise role of this circRNA on enzalutamide mechanism of action is under investigation at this point in time.”

  • Could you provide other models of enzalutamide resistance and find this marker again?

Different mechanisms of drug resistance have been identified (PMID: 27036029), therefore it is difficult to determine if this specific marker would be identified in other models. However, we appreciate that examining the expression of hsa_circ_0001275 in other models, would add value to our current data.

  • Minor revision: Please add STR profile of clone 1, 9 and parental LNCAP

We thank the reviewer for the suggestion and have now added the STR profile to the materials and methods in the manuscript page 3, line 108 to 113.

The amended text is as follows:

“The cell lines have undergone STR profiling and were compared against the ATCC STR database (https://www.atcc.org/search-str-database). Cell lines with ≥80% match are considered to be related; derived from a common ancestry. Using the following STR criteria: (AMEL) + the following loci: D5S818, D13S317, D7S820, D16S539, vWA, TH01, TPOX, CSF1PO. The results for identity using the ATCC algorithm were as follows: LNCAP Control: 94% (Pass), LNCaP C1: 94% (Pass), LNCaP C9: 94% (Pass).”

  • Please add starting plasma volume and the RNA quantification method applied

We have now added the starting plasma volume and RNA quantification method suggested by the reviewer to the materials and methods in the manuscript page 3, line 117 to 119.

The amended text is as follows:

“RNA was isolated from 200 µL plasma using the miRNeasy Plasma/Serum Kit (Qiagen) according to manufacturer’s instructions. The NanoDrop 1000 spectrophotometer (Thermo Fisher Scientific, MA, USA) was used to quantify the RNA.”